# Integrating multiple genomic technologies to investigate an outbreak of carbapenemase-producing *Enterobacter hormaechei*

Leah W. Roberts[1,2,3], Patrick N.A. Harris [4,5]*, Brian M. Forde [1,2,3], Nouri L. Ben Zakour[1,2,3], Elizabeth Catchpoole[5], Mitchell Stanton-Cook[1,2], Minh-Duy Phan [1,2], Hanna E. Sidjabat [2,4], Haakon Bergh[5], Claire Heney [5], Jayde A. Gawthorne [1,2], Jeffrey Lipman[4,6], Anthony Allworth[7], Kok-Gan Chan [8,9], Teik Min Chong[8], Wai-Fong Yin[8], Mark A. Schembri [1,2], David L. Paterson[2,4] & Scott A. Beatson [1,2,3]*

Carbapenem-resistant Enterobacteriaceae (CRE) represent an urgent threat to human health. Here we report the application of several complementary whole-genome sequencing (WGS) technologies to characterise a hospital outbreak of $bla_{IMP-4}$ carbapenemase-producing *E. hormaechei*. Using Illumina sequencing, we determined that all outbreak strains were sequence type 90 (ST90) and near-identical. Comparison to publicly available data linked all outbreak isolates to a 2013 isolate from the same ward, suggesting an environmental source in the hospital. Using Pacific Biosciences sequencing, we resolved the complete context of the $bla_{IMP-4}$ gene on a large IncHI2 plasmid carried by all IMP-4-producing strains across different hospitals. Shotgun metagenomic sequencing of environmental samples also found evidence of ST90 *E. hormaechei* and the IncHI2 plasmid within the hospital plumbing. Finally, Oxford Nanopore sequencing rapidly resolved the true relationship of subsequent isolates to the initial outbreak. Overall, our strategic application of three WGS technologies provided an in-depth analysis of the outbreak.

[1] School of Chemistry and Molecular Biosciences, The University of Queensland, Brisbane, QLD, Australia. [2] Australian Infectious Diseases Research Centre, The University of Queensland, Brisbane, QLD, Australia. [3] Australian Centre for Ecogenomics, The University of Queensland, Brisbane, QLD, Australia. [4] UQ Centre for Clinical Research, The University of Queensland, Brisbane, QLD, Australia. [5] Pathology Queensland, Central Microbiology, Brisbane, QLD, Australia. [6] Burns Trauma and Critical Care Research Centre, The University of Queensland, Brisbane, QLD, Australia. [7] Infectious Disease Unit, Royal Brisbane & Women's Hospital, Brisbane, QLD, Australia. [8] Division of Genetics and Molecular Biology, Institute of Biological Sciences, Faculty of Science, University of Malaya, Kuala Lumpur, Malaysia. [9] International Genome Centre, Jiangsu University, Zhenjiang, China. *email: p.harris@uq.edu.au; s.beatson@uq.edu.au

Carbapenem antibiotics have become the mainstay of therapy for serious infections caused by multidrug-resistant (MDR) gram-negative bacteria, especially for strains expressing extended-spectrum β-lactamase (ESBL) or AmpC-type enzymes[1]. Increased use has driven resistance to carbapenems and the emergence of carbapenemase-producing Enterobacteriaceae (CPE) and carbapenem-resistant Enterobacteriaceae (CRE), which include common enteric species such as Escherichia coli, Klebsiella pneumoniae and Enterobacter spp.[2].

Before 2005, an estimated 99.9% of Enterobacteriaceae were susceptible to carbapenems[3]. However, the isolation of CRE has since increased dramatically and these organisms are now reported in all WHO health regions[4]. The mortality rates for CRE infections are reported to be as high as 48%[5], and resistance to last-line antibiotics used in lieu of carbapenems, such as colistin, has also emerged[6].

Resistance to carbapenems in Enterobacteriaceae occurs via a range of mechanisms. Of greatest concern is the acquisition of genes encoding carbapenemases[7]. This most frequently occurs via transfer of mobile genetic elements (MGEs), such as plasmids, occasionally carrying multiple β-lactamases co-located with other resistance determinants, rendering these strains MDR or extensively drug-resistant (XDR)[8]. Australia has experienced low rates of CRE[9], although sporadic introduction of K. pneumoniae carbapenemase (KPC)[10] and New Delhi metallo-β-lactamase (NDM)[11] has been reported, including significant nosocomial outbreaks[12]. The most frequently encountered carbapenemase in Australia is $bla_{IMP-4}$, particularly in Enterobacter spp.[13]. IMP-producing Enterobacter spp. have caused occasional outbreaks within intensive care or burns units in Australian hospitals[14–16].

Within the last decade, whole-genome sequencing (WGS) has become more accessible and affordable, resulting in its increased use in many fields, including clinical microbiology. One of the main features of using WGS to characterise clinically relevant bacteria comes from its ability to provide strain relatedness at the resolution of a single nucleotide. WGS has been utilised to understand transmission within hospital settings beyond what can be determined using traditional culture-based diagnostics alone. Examples include the expansion of a single clone[17–19] and less commonly the expansion of an MGE[20,21]. However, many of these studies were conducted retrospectively and focused primarily on a single sequencing technology.

Here, we describe the integration of several WGS technologies to investigate an outbreak of IMP-4-producing Enterobacter hormaechei within an intensive care unit (ICU) and burns facility of a large tertiary referral hospital. Using Illumina sequencing, we were able to accurately determine the exact relationship between all outbreak isolates. Interrogation of publicly available data linked the outbreak strain to a 2013 isolate from the same ward, suggesting persistence of the organism in the environment for at least 2 years. Pacific Biosciences long-read sequencing enabled the detailed resolution of the $bla_{IMP-4}$ gene on a large IncHI2 plasmid, which we detected in seven unrelated IMP-4-positive Enterobacteriaceae from surrounding hospitals. Metagenomic sequencing of the hospital environment suggested the presence of both the outbreak strain and the IncHI2 plasmid in the hospital plumbing. We also applied Nanopore sequencing to reveal the true relationship of isolates subsequent to the original outbreak. Together, these results demonstrate the breadth of analyses that can be undertaken using a variety of sequencing techniques to completely characterise and monitor an outbreak.

## Results

### Clinical case report
Two patients in mid-2015 were transferred from regional Queensland hospitals to the ICU with burn injuries sustained from the same accident (Fig. 1). E. cloacae complex was cultured from the endotracheal tube (ETT) of patients 1 and 2 on days 6 and 8 of admission, respectively. Both E. cloacae complex isolates were confirmed as MDR by phenotypic testing used in the diagnostic setting (Table 1). Real-time PCR amplification of $bla_{IMP-4}$ confirmed their status as carbapenemase-producers. Both of these patients were previously well, with no prior hospital admission or contact with healthcare facilities. Neither had been resident or hospitalised overseas for more than 20 years.

Patient 1 underwent debridement and split skin grafting for 29% total body surface area burns on day 2 of ICU admission and subsequently had three procedures in the burns operating rooms (Fig. 1). An additional MDR-E. cloacae complex isolate was isolated from urine on day 21, 8 days after discharge from the ICU. After no further colonisation with MDR-E. cloacae complex was identified, patient 1 was discharged from the hospital on day 38.

Patient 2 underwent multiple grafting and debridement procedures and was discharged from the ICU on day 17 (Fig. 1). MDR-E. cloacae complex colonisation in samples from the ETT and from urine was noted on day 8 and day 15, respectively. By day 19, the patient developed clinical signs of sepsis, with a phenotypically identical isolate identified in blood cultures and from a central venous line (CVL) tip culture. The patient received piperacillin/tazobactam 4.5 g 8-hourly for 2 days, improved following line removal, and did not receive further antibiotics for this episode. A subsequent E. cloacae complex isolate in urine collected from a urinary catheter 17 days later demonstrated a different antibiogram with susceptibility to third-generation cephalosporins, meropenem and gentamicin. The patient received 3 days of oral norfloxacin 400 mg twice daily with microbiological resolution.

Patient 3 was admitted with 66% total body surface area burns to the same ICU 5 weeks after patients 1 and 2 were admitted and 20 days after they had been discharged from the ICU (Fig. 1). MDR-E. cloacae complex was cultured from the ETT of patient 3 on day 12 of ICU admission. The patient had frequent brief admissions to several hospitals since 2010 (never to ICU), and no MDR gram-negative bacilli were identified in clinical or screening samples during previous admissions. MDR-E. cloacae complex with Pseudomonas aeruginosa were isolated from eight skin swabs and an additional ETT aspirate. On days 19 and 21, MDR-E. cloacae complex was isolated from blood cultures in the context of skin graft breakdown and signs of systemic inflammatory response syndrome (SIRS) with increasing inotrope requirements (Fig. 1). Streptococcus mitis was cultured from blood on day 19. On day 36, the patient's condition worsened with signs of SIRS. Transesophageal echocardiography demonstrated aortic and mitral valve lesions consistent with endocarditis. Pancytopenia developed, with a bone marrow aspirate and trephine suggestive of peripheral consumption. Multiple suspected cerebral, pulmonary, splenic and renal septic emboli were identified on imaging. The patient was palliated on day 47 of admission due to extensive cerebral emboli (Fig. 1).

### All patients harbour carbapenemase-producing Enterobacter
With the exception of MS7889 (isolated from the urine of patient 2 on day 36), all E. cloacae complex isolates collected from the outbreak were resistant to ceftriaxone, ceftazidime, ticarcillin-clavulanate, piperacillin-tazobactam, meropenem, gentamicin and trimethoprim-sulphamethoxazole by Vitek 2 testing (Table 1) and demonstrated carbapenemase production by Carba-NP. The MICs for meropenem were considerably lower when tested by Etest[22], often falling below the clinically susceptible breakpoint defined by EUCAST, but above the

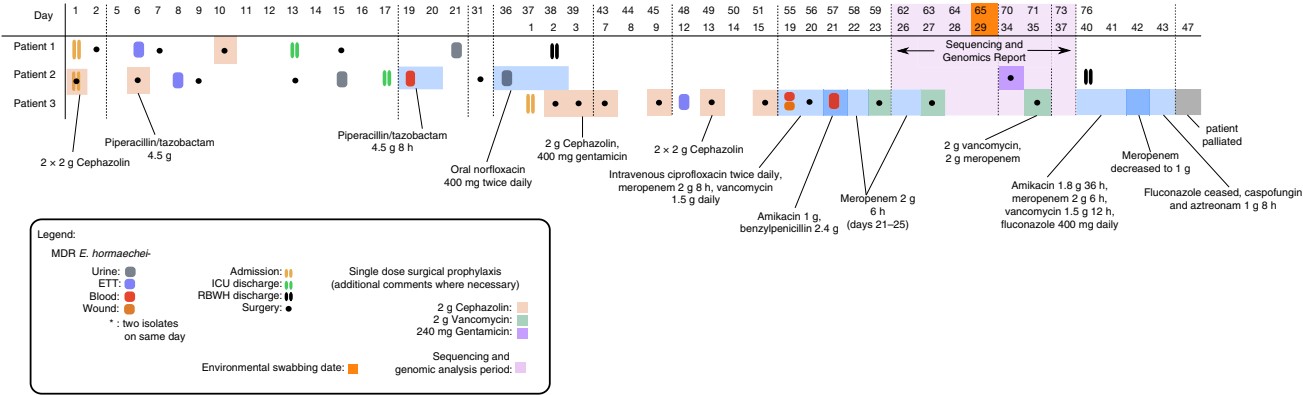

**Fig. 1 RBWH clinical case study outline.** Three burns patients were admitted to the RBWH ICU ward in mid-2015. Patients 1 and 2 were admitted on the same day. Subsequent to admission, both patients developed carbapenem-resistant *E. hormaechei* infections, with two samples taken from patient 1 (source = ETT [purple] and urine [grey]), and four samples taken from patient 2 (source = ETT [purple], urine [grey], and blood [red]). Patient 3 was admitted 37 days after patients 1 and 2 had been admitted and after they had been discharged from the ICU. Patient 3 also developed infection due to a carbapenem-resistant *E. cloacae* infection, and had four samples taken from ETT (purple), blood (red) and wound sites (orange). After intensive antibiotic and antifungal treatment, the patient was palliated on day 47 of ICU admission. Sequencing and genomics analysis of all ten isolates was undertaken following confirmation of all three patients being infected with $bla_{IMP-4}$-producing *E. hormaechei* (period shown in purple shading). Environmental swabbing was undertaken 65 days after the initial admission of patients 1 and 2, and 29 days after the admission of patient 3 (orange square).

epidemiological cut-off (ECOFF)[23]. MS7889 was fully susceptible to carbapenems (meropenem MIC = 0.032 by Etest) and was negative for IMP-4-like genes by PCR (Table 1).

**WGS identifies link to previous IMP4-producing isolate.** WGS of ten isolates from patients 1, 2 and 3 was initiated after an additional microbiological confirmation of a $bla_{IMP-4}$ *E. cloacae* complex isolate from a third patient from the RBWH ICU (Fig. 1 and Supplementary Table 1). In silico analysis determined all to be sequence type (ST)90 *E. hormaechei* (part of the *E. cloacae* complex), with the majority exhibiting the same resistance gene profile, including a 100% identical $bla_{IMP-4}$ gene (Table 1). The exception was the carbapenem-susceptible isolate MS7889, which was confirmed by WGS to have lost the $bla_{IMP-4}$ gene as well as several additional resistance genes conserved in the other *E. hormaechei* isolates (Table 1). All ten isolates contained an IncHI2 plasmid. Sequence analysis suggested that AmpC derepression was unlikely to contribute to carbapenemase activity in these strains (see Supplementary Note 1 for further details).

Comparison of the *E. hormaechei* genomes to publicly available draft assemblies identified a close match to *E. hormaechei* Ecl1 (GenBank: JRFQ01000000; formerly *E. cloacae*), an ST90 strain isolated from a burns patient at the RBWH ICU almost 2 years prior to the 2015 outbreak[13,24]. Antibiotic resistance gene profiling of the Ecl1 genome revealed an identical resistance gene profile compared to the majority of the 2015 isolates (Table 1).

**The 2015 isolates were near-identical to a 2013 isolate.** To investigate the relationship between the isolates at single-nucleotide resolution, reads from the 2015 RBWH isolates were mapped to *E. hormaechei* draft assembly for Ecl1. All 2015 RBWH isolates differed by fewer than five core single nucleotide variants (SNVs) (4,934,357 bp core genome), consistent with a direct ancestral relationship (Fig. 2 and see Supplementary Note 2). Two isolates from patient 1 and two isolates from patient 3 were indistinguishable at the core genome level (Fig. 2), although all of the isolates from patient 3 had lost a prophage region (see Supplementary Note 3 and Supplementary Fig. 1). Ecl1 (isolated in 2013) was very closely related to these isolates, differing by only one core SNP. All four isolates from patient 2

contained a discriminatory single-nucleotide deletion, thereby ruling out patient 2 to patient 3 transmission (Fig. 2).

**Integration of WGS with infection control response.** WGS analysis unequivocally linked all ten isolates to the 2013 isolate Ecl1 from the same ward, confirming that the clone had not been an incursion from the accident affecting patients 1 and 2 and that the hospital environment was suspected as the most likely original source of infection in the 2015 cases. In response, 28 environmental samples from the ICU, burns wards and operating theatres were collected 65 days after patients 1 and 2 were admitted and inoculated onto MacConkey agar with 8 mg/mL gentamicin (laboratory standard screening medium for MDR gram-negative bacilli). No carbapenemase-producing *Enterobacter* spp. were detected. Additionally, no carbapenemase-producing *Enterobacter* spp. were detected in patients admitted to the ICU or burns unit for a 6-month period following the outbreak.

**Queensland CPE isolates associated with an IMP-4-carrying plasmid.** To determine the broader context of IMP-producing Enterobacteriaceae in surrounding hospitals, seven additional $bla_{IMP-4}$-producing Enterobacteriaceae (*E. cloacae* complex n = 6, *E. coli* n = 1) were sequenced. These represented a selection of $bla_{IMP-4}$-producing Enterobacteriaceae identified from Brisbane public hospitals via Pathology Queensland Central Microbiology for 2015. Both MLST and SNP analysis found no relationship to the 2015 RBWH *E. hormaechei* isolates, with approximately 50,000 SNP differences between the ST90 representative strain Ecl1 and its nearest non-ST90 phylogenetic neighbour (Fig. 3). Despite not being clonally related, all additional Enterobacteriaceae isolates possessed very similar antibiotic resistance gene profiles (Supplementary Table 2), suggesting the possibility of lateral gene transfer via MGEs (e.g. integrons and/or plasmids). WGS analysis revealed that all 18 CPE isolates in this study, including the *E. coli* isolate, harboured an IncHI2 plasmid (plasmid ST1) and an identical $bla_{IMP-4}$ gene, strongly suggesting plasmid-mediated circulation of $bla_{IMP-4}$ between Enterobacteriaceae in Brisbane hospitals.

**Table 1 Antibiotic resistance profile as determined by Etest, Vitek 2 and ResFinder**

| Patient | | | 1 | 2 | | | | | 3 | | | |
|---|---|---|---|---|---|---|---|---|---|---|---|---|
| Strain (MS) | | | 7884 | 7885 | 7886 | 7887 | 7888 | 7889 | 7890 | 7891 | 7892 | 7893 |
| Source | | | ETT | urine | ETT | urine | blood | urine | ETT | blood | Leg swab | blood |
| Carbapenems | E-test | Ertapenem | 1 | 2 | 4 | 2 | 0.5 | 0.032 | 2 | 0.5 | 0.5 | 2 |
| | | Imipenem | 2 | 1 | 4 | 8 | 1 | 0.5 | 2 | 1 | 1 | 4 |
| | | Meropenem | 0.5 | 1 | 4 | 2 | 0.5 | 0.032 | 2 | 1 | 0.5 | 0.5 |
| β-lactams and Cephalosporins | Vitek2 | Tim | ≥128 | ≥128 | ≥128 | ≥128 | ≥128 | 32 | ≥128 | ≥128 | ≥128 | ≥128 |
| | | Mer | ≥16 | ≥16 | ≥16 | ≥16 | ≥16 | ≤0.25 | ≥16 | ≥16 | ≥16 | ≥16 |
| | | Taz | 16 | 16 | 16 | 16 | 16 | 8 | 16 | 16 | 16 | 16 |
| | | Fox | ≥64 | ≥64 | ≥64 | ≥64 | ≥64 | ≥64 | ≥64 | ≥64 | ≥64 | ≥64 |
| | | Caz | ≥64 | ≥64 | ≥64 | ≥64 | ≥64 | ≤1 | ≥64 | ≥64 | ≥64 | ≥64 |
| | | Cro | 16 | 16 | 16 | 16 | 16 | ≤1 | 16 | 16 | 16 | 8 |
| | | Fep | 2 | 2 | 4 | 2 | 2 | ≤1 | 2 | 2 | 2 | 4 |
| | Res | ampC | + | + | + | + | + | + | + | + | + | + |
| | | bla$_{OXA-1}$ | + | + | + | + | + | + | + | + | + | + |
| | | bla$_{IMP-4}$ | + | + | + | + | + | − | + | + | + | + |
| | | bla$_{TEM-1B}$ | + | + | + | + | + | + | + | + | + | + |
| Aminoglycosides | Vitek2 | Ami | ≤2 | ≤2 | ≤2 | ≤2 | ≤2 | 8 | ≤2 | ≤2 | ≤2 | ≤2 |
| | | Gent | ≥16 | ≥16 | ≥16 | ≥16 | ≥16 | ≤1 | ≥16 | ≥16 | ≥16 | ≥16 |
| | | Tob | 8 | 8 | 8 | 8 | 8 | ≥16 | 8 | 8 | 8 | 8 |
| | Res | strB | + | + | + | + | + | + | + | + | + | + |
| | | strA | + | + | + | + | + | + | + | + | + | + |
| | | aac(6')Ib-cr | + | + | + | + | + | + | + | + | + | + |
| | | aac(3)-IId | + | + | + | + | + | − | + | + | + | + |
| Quinolones | Vitek2 | Cip | ≤0.25 | 0.5 | ≤0.25 | ≤0.25 | 0.5 | ≤0.25 | 0.5 | 0.5 | 1 | ≤0.25 |
| | | Nor | 2 | 2 | 2 | 2 | 2 | 0.5 | 2 | 2 | 2 | 1 |
| | Res | qnrB2 | + | + | + | + | + | − | + | + | + | + |
| Sulphonamide/Trimethoprim | Vitek2 | Tmp/smx | ≥320 | ≥320 | ≥320 | ≥320 | ≥320 | ≥320 | ≥320 | ≥320 | ≥320 | ≥320 |
| | Res | sulI | + | + | + | + | + | + | + | + | + | + |
| Rifampicin | Res | dfrA18 | + | + | + | + | + | − | + | + | + | + |
| | Res | arr3 | + | + | + | + | + | − | + | + | + | + |
| Macrolide | Res | mph(A) | + | + | + | + | + | − | + | + | + | + |
| Phenicols | Res | catA2 | + | + | + | + | + | + | + | + | + | + |
| | Res | catB3 | + | + | + | + | + | + | + | + | + | + |
| Tetracycline | Res | tet(D) | + | + | + | + | + | + | + | + | + | + |

E-test E-test MIC (mg/L), Res ResFinder Antimicrobial Resistance gene database, Vitek2 Vitek 2 automated susceptibility MIC (mg/L), Tim ticarcillin–clavulanate, Taz piperacillin–tazobactam, Fox cefoxitin, Caz ceftazidime, Cro ceftriaxone, Fep cefepime, Mer meropenem, Ami amikacin, Gent gentamicin, Tob tobramycin, Cip ciprofloxacin, Nor norfloxacin, Tmp/smx trimethoprim–sulphamethoxazole

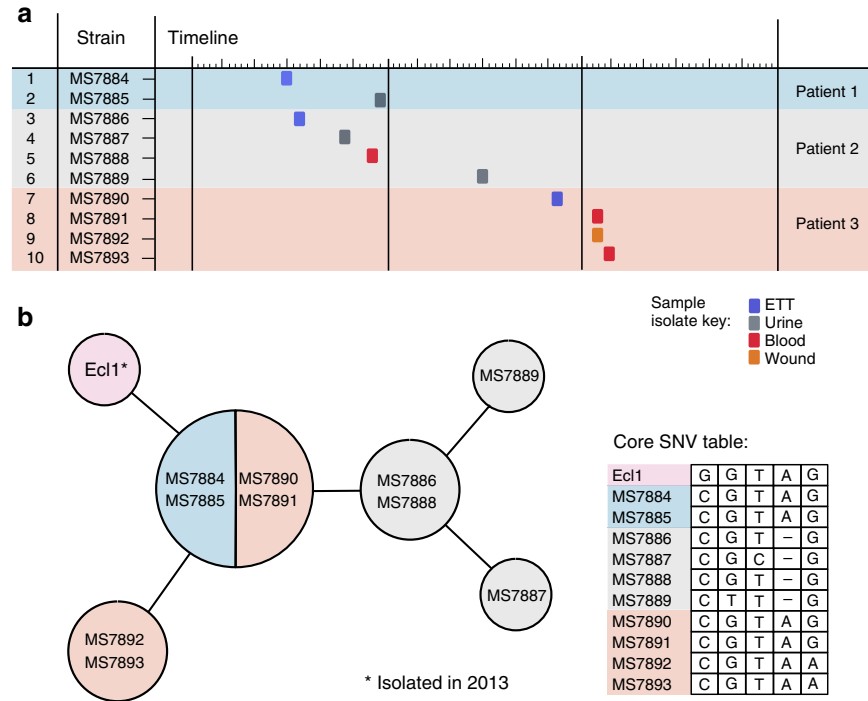

**Fig. 2 CPE isolate timeline and relationship matrix. a** Ten isolates were collected from three patients at various time-points in mid-2015. Coloured blocks indicate the source of the isolated strain: purple: respiratory, grey: urine, red: blood, and orange: wound. **b** Relationship matrix (left) shows specific core single nucleotide variant (SNV) differences identified between strains. Strains within the same circle have identical core SNV profiles. Lines connecting circles represent accumulating SNV differences between strains (not-to-scale), where each line represents one SNV (including nucleotide deletion). Specific nucleotide differences between isolates are given in the table in panel (**b**). Locations and consequences of nucleotide change are shown in Supplementary Data 1. All 11 isolates differed by 5 SNVs overall.

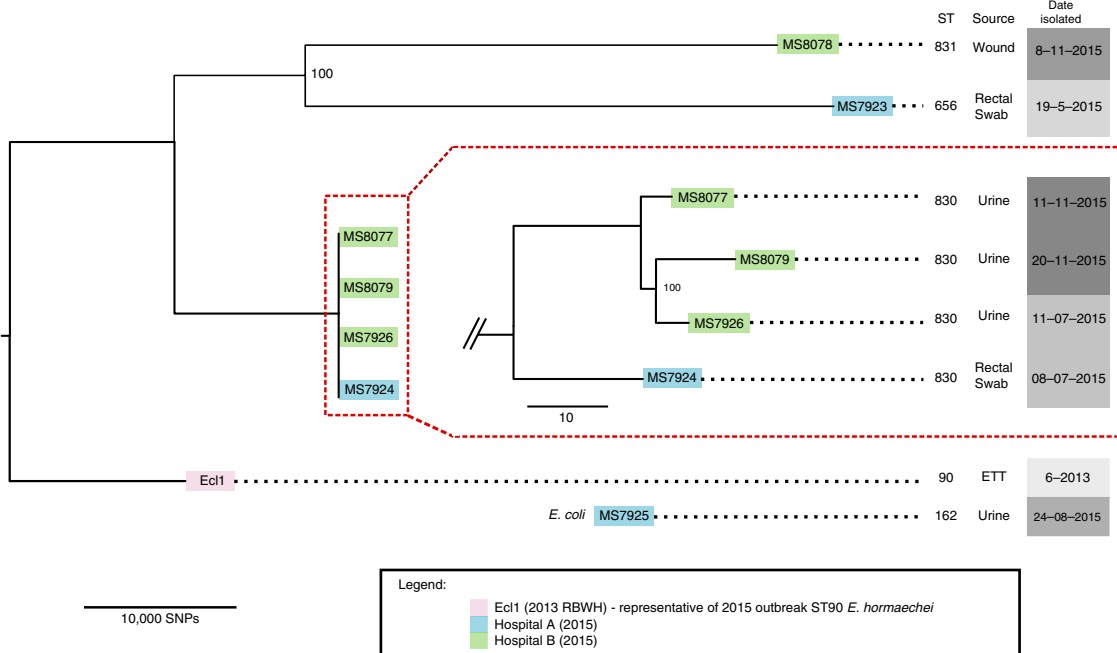

**Fig. 3 *E. hormaechei* isolates from Hospitals A and B in relation to RBWH isolates.** Trimmed reads from six *E. hormaechei* isolates (Hospitals A and B) were aligned to the reference *E. hormaechei* Ecl1 (isolated in 2013 at the RBWH) to determine core single nucleotide polymorphisms (SNPs) between all isolates. Ecl1 in this figure represents all 2015 RBWH isolates (*n* = 10) as they were found to be near-identical at the core genome level. 63,861 core SNPs were identified and used to generate an ML tree with RAxML (1000 bootstrap replicates), which determined no relationship between the RBWH isolates (pink) and the Hospital A (blue)/Hospital B (green) isolates. Four closely related strains were identified from Hospitals A and B (red box). Alignment of trimmed reads from MS8077, MS8079 and MS7926 to MS7924 identified 117 core SNPs; however, a number of these SNPs were removed as they were identified as residing within transposon or phage regions. The remaining 58 core SNPs were used to generate an ML tree (1000 bootstrap replicates), showing that Hospital B strains differ by less than 20 SNPs.

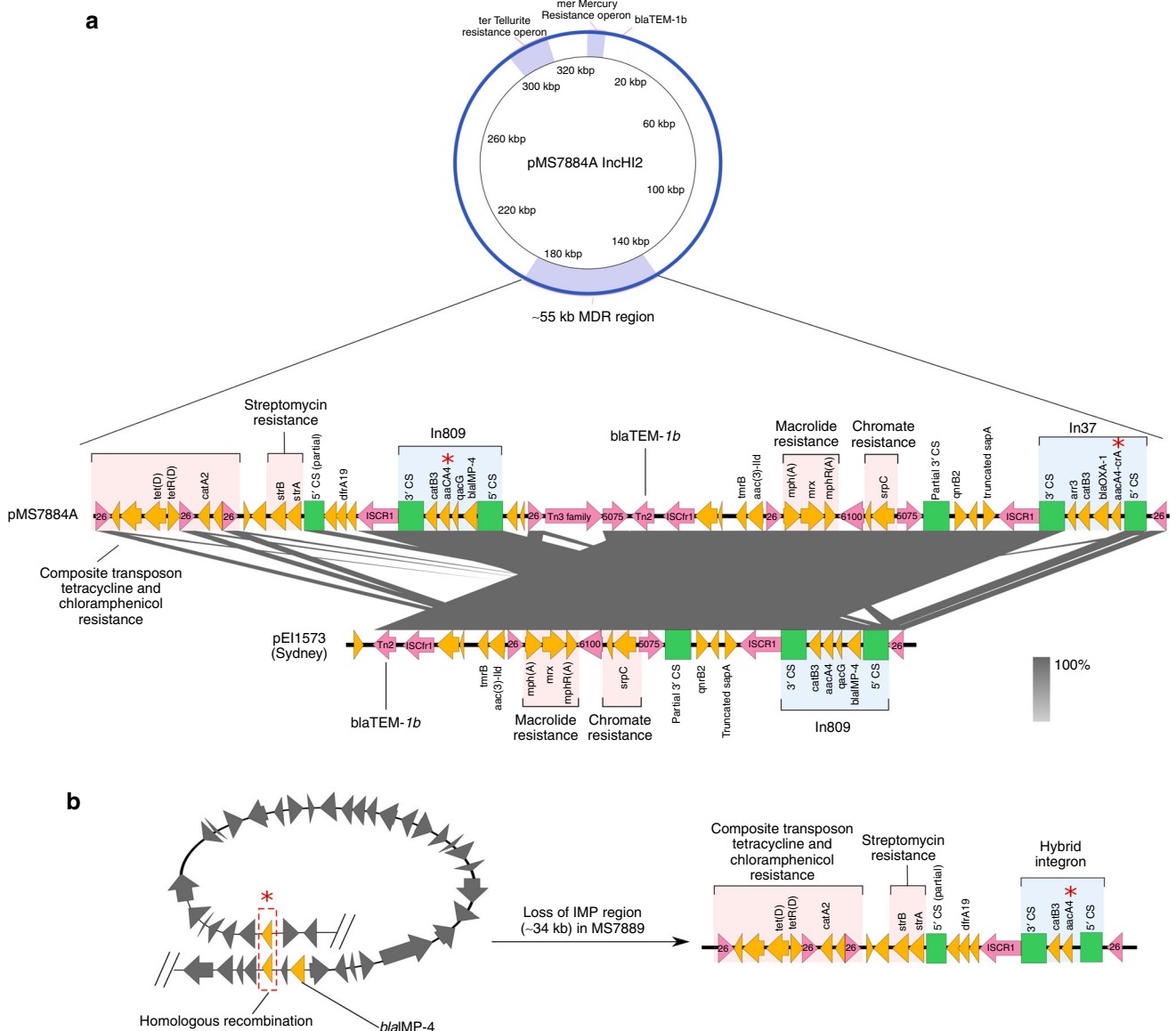

**Fig. 4 Large IncHI2 plasmid with ~55 kb multidrug resistance region containing bla_IMP-4.** **a** A 330,060 bp IncHI2 plasmid carrying multiple resistance operons, including a large ~55 kb multidrug resistance (MDR) region, was fully recovered and assembled using Pacific Biosciences (PacBio) SMRT sequencing of strain MS7884 (patient 1, isolate 1). The multidrug resistance region was found to contain two class 1 integrons (In809, In37) along with several other antibiotic resistance genes, as indicated. Comparison of this MDR region to publicly available genomes found a close match to pEl1573, isolated in 2012 from an *E. cloacae* isolate in Sydney, Australia. **b** A predicted model of homologous recombination between two nearly identical *aacA4/aac (6′)-Ib-cr* genes (red asterisks) within the ~55 kb MDR region in MS7889 (patient 2, isolate 4, IMP-, carbapenem-susceptible) leading to the loss of a ~34 kb region containing *bla*_IMP-4 as well as several other antibiotic resistance genes.

**The *bla*_IMP-4 gene resides on an IncHI2 plasmid.** Due to the presence of multiple repetitive elements surrounding *bla*_IMP-4, including insertion sequences (IS) and two suspected integrons with similar gene content, we were unable to accurately resolve the context of *bla*_IMP-4 using Illumina sequencing alone. One representative isolate (MS7884) was sequenced twice using Pac-Bio SMRT sequencing (see Supplementary Notes 4 and 5), which was able to resolve a complete closed chromosome of 4,810,853 bp and two plasmids (Supplementary Table 3 and Supplementary Note 6): pMS7884A, a 330,060 bp IncHI2 plasmid carrying *bla*_IMP-4 within a ~55 kb MDR region (Fig. 4a, Supplementary Fig. 2), and pMS7884B, a smaller untypeable plasmid of 126,208 bp (Supplementary Fig. 3). The pMS7884A MDR region harbours two different class 1 integrons (In37 and In809) as well as a composite transposon conferring resistance to tetracycline

and chloramphenicol (Fig. 4a). BLASTn and read-mapping analysis revealed the presence of identical plasmids in all but one of the 18 isolates sequenced by Illumina in this study: isolate MS7889 is predicted to have lost a ~34 kb region from its MDR plasmid, including *bla*_IMP-4, due to homologous recombination between two almost identical aminoglycoside resistance genes (Fig. 4b, Supplementary Notes 7 and 8, Supplementary Figs. 4 and 5). Notably in 15% of cases, sub-culture of MS7884 in the absence of meropenem selection resulted in loss of *bla*_IMP-4 or the entire plasmid (Supplementary Note 9).

**Continued WGS surveillance reveals persistence of outbreak.** Ongoing surveillance has been in place for *bla*_IMP-4-positive *E. hormaechei* isolates within the hospital since 2015. In 2016,

during an unrelated outbreak, a $bla_{IMP-4}$-positive *E. hormaechei* was isolated (MS14389) and following Illumina sequencing found to be only two SNPs different from the 2015 isolates, demonstrating continued persistence in the hospital environment (Supplementary Fig. 6).

In 2017, another $bla_{IMP-4}$-positive *E. hormaechei* (MS14449) was isolated from the haematology ward. Oxford Nanopore MinION sequencing has advanced in recent years to provide long-read sequencing and real-time analysis of bacterial isolates[19]. However, it remains a challenge to rapidly contextualise new isolates during an outbreak, in part due to the relatively high single-read error rate of long-read data. These errors, which are not easily differentiated from true differences, make it difficult to determine the precise genetic relationship between new isolates and the outbreak strain. However, we found that Nanopore data alone was sufficient to rule out MS14449 from the ongoing outbreak within 2 days of receiving the isolate (see Supplementary Note 10). By contextualising the de novo Nanopore assembly with genomes of publicly available *E. cloacae* complex strains and existing outbreak isolates, we determined that the *E. hormaechei* isolated was not related to the 2015 outbreak isolates, but did carry a near-identical IncHI2 plasmid (Supplementary Figs. 7–9). A $bla_{IMP-4}$-positive *K. pneumoniae* isolate (MS14448) taken from the same patient was also sequenced (using only Illumina short-read sequencing) and found to carry a very similar IncHI2 plasmid, albeit missing a small section of the MDR region (Supplementary Fig. 10).

**Shotgun metagenomic sequencing reveals source in plumbing**. Despite continued routine surveillance of the hospital environment using traditional culture methods, an environmental source for the ST90 *E. hormaechei* was not found. In July 2018, 50 swab and water samples from the ICU and Burns ward environments were collected in response to an unrelated outbreak and subjected to both shotgun metagenomic sequencing and traditional culturing (Supplementary Data 2). From this round of surveillance, *Klebsiella oxytoca*, *E. cloacae* complex and *Leclercia adecarboxylata* were detected via traditional culturing methods from four samples (Supplementary Table 4); however, Illumina sequencing of these isolates determined that they were unrelated to the outbreak. Despite being clonally unrelated, three of these isolates were found to carry a $bla_{IMP-4}$-like gene (based on real-time PCR), which upon further inspection of the sequencing data corresponded to an IncHI2 plasmid with high identity to pMS7884A (Supplementary Fig. 11).

While traditional culturing was unable to detect the ST90 *E. hormaechei*, metagenomic sequencing identified two samples with high confidence matches to the ST90 *E. hormaechei* reference MS7884 and the IncHI2 plasmid pMS7884A. Nucleotide comparison of the metagenomic assembled genomes (MAGs) for these samples (R5514 and R5537, both taken from floor drains) to our *E. hormaechei* reference genome revealed a high level of nucleotide identity across the entirety of the chromosome and plasmid, equivalent to 93−97% average nucleotide identity (ANI) across >95% of the reference sequences (Supplementary Data 2 and Supplementary Figs. 12 and 13). MLST analysis of both MAGs was also able to detect an almost identical set of alleles for ST90 (Supplementary Table 5). Additionally, screening for resistance genes identified the $bla_{IMP-4}$ gene in both samples, further supporting the presence of ST90 *E. hormaechei* and an IncHI2 plasmid similar to pMS7884A in the floor drains. Analysis of three other environmental samples (R5505, R5506 and R5522) revealed low confidence hits to *E. hormaechei* (likely non-ST90) and/or the MDR region in pMS7884A (Supplementary Table 5). MAGs for these samples were unable to be matched with

sufficient sensitivity to our reference genome to the extent that the strain could be confidently identified (Supplementary Table 5 and Supplementary Figs. 12 and 13).

## Discussion

While there has been a dramatic improvement in the cost and availability of WGS, incorporating these advances into routine clinical microbiology remains a challenge. Several studies have demonstrated the ability of WGS to provide optimal discrimination between strains to help inform a response to outbreaks or nosocomial acquisition[17,25–27]. Here, we demonstrate that integrated WGS approach can help rapidly characterise an outbreak in a critical care setting, particularly regarding transmission pathways. We highlight how WGS can be used to link contemporary outbreak isolates to historical isolates to inform infection control and incorporate long-read sequencing technologies to resolve complete genomes (including plasmids) and to rapidly resolve suspected outbreak cases. Finally, we demonstrate the potential of using complete genome sequences to interrogate environmental shotgun metagenomic sequencing data to identify outbreak sources.

The finding that the outbreak strains were virtually indistinguishable from an IMP-4-producing *E. hormaechei* isolate identified 2 years previously from the same unit was unexpected and highlighted the need to consider both environmental sources and potential person-to-person transmission during outbreak investigation, as has been previously described in Australian ICU and burns units[14]. Despite ongoing surveillance, traditional culture-based detection methods were unable to find the ST90 *E. hormaechei* in the environment. Direct DNA extraction and metagenomic sequencing has in recent years revolutionised infectious disease surveillance, allowing detection of all microorganisms in a sample without the biases and limitations of traditional pathogen detection[28]. Using metagenomic sequencing and a high-quality complete reference genome we were able to detect two samples with high confidence hits to ST90 *E. hormaechei* and an IncHI2 plasmid, confirming its presence in the environment. We were also able to observe the overall community profile within each environmental sample, supporting the use of metagenomic sequencing as a powerful infection control and surveillance tool for tracking (i) the types of bacteria present in the environment, (ii) the types of resistance genes circulating, and (iii) the effectiveness of environmental cleaning. Metagenomic sequencing does, however, have certain limitations when considering its implementation in routine infection control, including the necessity for a reasonable amount of starting DNA, and the chance of amplification inhibition during library preparation (causing low yield or failed sequencing). Metagenomic sequencing can also be quite costly[29], as the sequencing output needs to be sufficiently high to provide an accurate population structure and detect low-abundance organisms. In our study, while metagenomic sequencing of the environmental samples yielded positive results, it is still unclear how these reservoirs might act as a source for reinfection in patients. It is possible that healthcare workers are somehow involved, with previous studies confirming carriage of a range of clinically important bacteria[30–32].

Using SMRT sequencing technology, we determined the full context of $bla_{IMP-4}$ and its location within a large, complex and highly repetitive MDR region harbouring two integrons: In37 and In809. In37 is a widespread class 1 integron that has been found in many bacterial species[33,34]. In809, which carries $bla_{IMP-4}$, has previously been described from *K. pneumoniae* (GenBank: KF250428.1, HQ419285.1, AJ609296.3), *E. cloacae* (GenBank: JX101693.1) and *Acinetobacter baumannii* (GenBank: AF445082.1, DQ532122.1) in various plasmid backgrounds including

IncA/C2[35], IncL/M and IncF[36]. Most recently, a carbapenemase-producing *Salmonella* sp. isolated from a domestic cat in Australia was shown to contain $bla_{IMP-4}$ within an IncHI2 MDR plasmid (pIMP4-SEM1)[37]. Remarkably, we found that pIMP4-SEM1 was near-identical to pMS7884A (Supplementary Fig. 2). This finding supports a role for domestic animals in the spread of antibiotic resistance genes.

Analysis of several CPE in this study suggested that a common plasmid or integron carrying multiple antibiotic resistance genes is likely the major driver of antibiotic resistance dissemination across a broad range of Enterobacteriaceae. In addition to the presence of $bla_{IMP-4}$, four resistance genes ($bla_{TEM-1b}$, $bla_{IMP-4}$, $qnrB$, and $aac(6')$-$Ib$) carried by these isolates were previously detected by PCR in the majority of 29 IMP-4-producing *E. cloacae* complex isolates surveyed from Queensland hospitals between June 2009 and March 2014[13]. Only one of these isolates was ST90, suggesting lateral transfer of these genes to different *Enterobacter* clones in Queensland before 2013. During ongoing surveillance at RBWH we identified an *E. hormaechei* isolate that also carried the $bla_{IMP-4}$ plasmid, but was not closely related to the outbreak isolates, suggesting ongoing later transfer within the hospital environment. In this instance, the Oxford Nanopore MinION sequencing platform was instrumental in ruling out this isolate from the ongoing outbreak of $bla_{IMP-4}$ ST90 *E. hormaechei*. Our work highlights the potential for integrating this highly portable and rapid technology as part of the "genomic toolkit" for infection control alongside more established platforms.

There were significant discrepancies between meropenem MICs according to the testing modality used, with the Etest consistently testing as "susceptible/intermediate" (MIC ≤ 4 mg/L; range 0.5–4 mg/L) and Vitek 2 as "resistant" (usually with MICs ≥ 16 mg/L). According to pharmacokinetic/pharmacodynamic (PK/PD) principles, provided the MIC to a carbapenem falls within a susceptible range, the agent may still be effective despite the presence of a carbapenemase[38]. Robust clinical data to help guide therapy are lacking and many clinicians rely on combination therapy to optimise efficacy against carbapenemase-producers, largely based on observational studies suggesting benefit[39,40]. The presence of carbapenemase genes may be missed if clinical breakpoints for carbapenem MICs are used[23], however it can be rapidly ascertained by WGS, without a priori assumptions of which genes are likely to be present. A wealth of additional information that may influence clinical decisions can be obtained, such as the presence of other β-lactamases, factors that may regulate resistance gene expression (e.g. IS elements), mutations in outer-membrane proteins, or other known resistance genes.

We used an integrated WGS approach to help elucidate genetic relationships between $bla_{IMP-4}$ carbapenemase-producing *E. hormaechei* identified from our ICU and Burns facility. Rapid application of this technology revealed an unexpected clonal relationship with a strain isolated from the same unit 2 years previously. Continued routine WGS surveillance has enabled detailed monitoring of the outbreak, with rapid nanopore sequencing crucial for ruling out a suspected case in a previously unaffected ward. Comparisons with other Enterobacteriaceae containing $bla_{IMP-4}$ isolated from surrounding hospitals revealed its carriage on a broad host range IncHI2 plasmid, assumed to be circulating via lateral gene transfer across different *E. cloacae* complex clones and also *E. coli*. SMRT sequencing enabled the genetic context of all resistance genes within this plasmid to be resolved and revealed the mechanism of loss of resistance genes in one *E. hormaechei* strain that reverted to a fully carbapenem-susceptible phenotype. The availability of a complete *E. hormaechei* reference chromosome and $bla_{IMP-4}$ plasmid were also instrumental in locating a suspected source of the outbreak in the hospital plumbing with shotgun metagenome sequencing. As WGS technologies become increasingly available, they are likely to prove essential tools for the clinical microbiology laboratory to respond to emergent infection control threats, with advances in their real-time use to provide important and timely information for infection control personnel.

## Methods

**Study setting.** Primary isolates were obtained from patients admitted to the Royal Brisbane & Women's Hospital (RBWH), a tertiary referral hospital with 929 beds in South-East Queensland, Australia. Additional IMP-producing isolates, cultured from patients admitted to other hospitals in the metropolitan Brisbane area (referred to as Hospitals A and B), were obtained from Pathology Queensland—Central Microbiology for comparison (Supplementary Table 2).

**Ethics and consent.** Ethical approval for this work was provided by the Metro North Hospital and Health Service Human Research Ethics Committee (HREC/16/QRBW/253). In addition, written consent was obtained from the patients or their next of kin to publish clinical details.

**Antimicrobial susceptibility and carbapenemase detection.** All bacterial isolates were identified by matrix-assisted laser desorption/ionisation mass spectrometry (MALDI-TOF) (Vitek MS; bioMérieux, France). Antimicrobial susceptibility testing was carried out using Vitek 2 automated AST-N426 card (bioMérieux) with Etest to determine MICs for meropenem, imipenem and ertapenem. Carbapenemase activity was assessed by the use of the Carba-NP test (RAPIDEC; bioMérieux) and the presence of the $bla_{IMP-4-like}$ carbapenemase gene confirmed using an in-house multiplex real-time PCR (also targeting NDM, KPC, VIM and OXA-48-like carbapenemases)[41].

**Bacterial DNA extraction.** Single colonies were selected from primary bacterial cultures and grown in 10 mL Luria Bertani (LB) broth at 37 °C overnight (shaking 250 rpm). DNA was extracted using the UltraClean® Microbial DNA Isolation Kit (MO BIO Laboratories) as per the manufacturer's instructions.

**Genome sequencing, quality control and de novo assembly.** Illumina sequencing of the 2015 $bla$IMP + *E. hormaechei* isolates ($n = 16$) and *E. coli* isolate ($n = 1$) was done at the Australian Centre for Ecogenomics (ACE). *E. hormaechei* Ecl1 (formerly identified as *E. cloacae*) had previously been sequenced on Illumina HiSeq2000 at the Australian Genome Research Facility (Melbourne, Australia). Libraries for isolates MS7884-MS7893 (RBWH isolates) were prepared using 2 × 150 bp Nextera XT V1 chemistry and run on an Illumina NextSeq. Isolates MS7923-MS7926 and MS8077-MS8079 (Hospitals A and B) were prepared using 2 × 300 bp Nextera XT libraries with V3 chemistry and run on an Illumina MiSeq. All libraries were quantified using Viaa7 Thermo Fisher qPCR with QC checks with bioanalyzer analysis (Agilent). Contaminant searches on the raw reads were performed using Kraken (v0.10.5-beta). All raw reads were filtered using Nesoni (v0.130)[42] to remove Illumina adaptor sequences, reads shorter than 80 bp and bases below Phred quality 5. MiSeq reads were additionally hard trimmed to 150 bp due to low-quality bases between 1–10 and 160–300 using Nesoni (v0.130).

All isolates collected subsequent to the original outbreak were sequenced at Queensland Forensic Scientific Services (MS14449, MS14448, MS14389, M87132, M87133, M87134, and M87135). DNA was extracted using the DSP DNA Mini Kit on the QIAsymphony SP (Qiagen). Libraries were prepared using the Nextera XT DNA preparation kit (Illumina) and the sequencing was performed on the NextSeq 500 (Illumina) with 2 × 150 bp chemistry, NextSeq Midoutput kit v2.5.

Reads passing quality control (QC) were assembled using Spades v3.6.0[43] under default parameters (without careful flag). Contigs with coverage less than 10× were removed from final assemblies. Final assembly metrics were checked using QUAST v2.3[44] (Supplementary Table 1).

**Taxonomic identification.** Illumina raw reads were analysed using Kraken v0.10.5-beta to determine species and possible contamination. With the exception of MS7925, which was found to be *E. coli*, initial analysis determined the isolates to be *E. cloacae*. De novo assemblies for all *E. cloacae* complex isolates were compared to representative publicly available complete *E. cloacae* complex genomes using MASH v2.0[45] and FastANI v1.1[46] at default settings. Genome comparisons were made against *Enterobacter asburiae* (NZ_CP011863.1), *Enterobacter bugandensis* (NZ_LT992502.1), *Enterobacter cloacae* subsp. *cloacae* (NC_014121.1), *Enterobacter hormaechei* subsp. *steigerwaltii* (NZ_CP017179.1), *Enterobacter kobei* (NZ_CP017181.1), *Enterobacter ludwigii* (NZ_CP017279.1), *Enterobacter roggenkampii* (NZ_CP017184.1), *Enterobacter hormaechei* subsp. *xiangfangensis* (NZ_CP017183.1), and *Enterobacter hormaechei* subsp. *hoffmannii* (NZ_CP017186.1). The top hit, based on %ANI and greatest number of matching hashes (for fastANI and MASH respectively), was used to identify the species for that isolate.

**Determining reference for phylogenetic analysis.** Kraken (v0.10.5-beta) analysis using raw reads from the ten 2015 RBWH strains determined *E. cloacae* subsp. cloacae NCTC 9394 (GenBank: FP929040.1) to be the closest match in the complete genome division of GenBank. Subsequent comparison of the ten 2015 RBWH draft genome assemblies to assemblies within the publicly available whole-genome shotgun (WGS) database revealed a closer match to the IMP-4-producing *E. hormaechei* strain Ecl1 from 2013[24] (Accession: JRFQ01000001; formerly identified as *E. cloacae*). Both were subsequently used to confirm single nucleotide variations (SNVs) between the 2015 RBWH strains.

**Phylogenetic analysis.** SHRiMP v2.2.3[47] as implemented in Nesoni v0.130[42] under default settings was used to determine core single nucleotide polymorphisms (SNPs) between the ten 2015 RBWH *E. hormaechei* genomes to reference Ecl1 and create a minimum spanning tree (see Supplementary Note 11 for details on use of alternative bioinformatics tools). Maximum likelihood trees of Ecl1 and the six *E. cloacae* from Hospitals A and B were built using RAxML v8.1.15[48] based on the Nesoni core SNPs. RAxML was run with the GTRGAMMA nucleotide substitution rate and an initial seed length of 456 (bootstrap 1000 with Lewis ascertainment bias correction). Core genome size was estimated using Parsnp v1.2[49].

**In silico genotyping methods.** Multi-locus sequencing typing (MLST) of isolate raw reads was performed using srst2 v0.1.5[50] with typing schemes available on PubMLST[51]. Plasmid typing was done based on Compain et al.[52] by generating in silico PCR products from the reference strains used, and similarity searches using BLASTn v2.2.3. Antimicrobial (AMR) genes were detected using the ResFinder database[53] and the ARG-ANNOT database[54] with BLASTn v2.2.3 and srst2 v0.1.5[50] respectively. Manual confirmation was carried out using BLASTn and read mapping using Burrows–Wheeler Aligner (BWA v0.7.5a-r405)[55].

**Reassembly of Ecl1 draft genome from raw reads.** In order to determine the relationship between *E. hormaechei* Ecl1 and the 2015 RBWH *E. hormaechei* strains, core SNP distances were determined using a mapping approach as implemented through Nesoni v0.130[42] against the publicly available Ecl1 genome (Accession: JRFQ01000001). Manual inspection of the Nesoni output identified a number of clustered SNPs that we suspected were erroneous (Supplementary Data 1). Reassembly of the raw reads using Spades v3.6.0[43] under default parameters (without careful flag) was unable to correct the erroneous SNPs (Supplementary Data 1). Closer examination of the read pileups for Ecl1 identified strand-specific clusters of SNPs, possibly due to technical problems at the time of sequencing. To identify genuine variants, the read pileup for every SNP as reported by Nesoni was manually curated using Artemis and Bamview. SNPs that were due to strand-specific base call errors in the assembly, resided close to contig edges, or resided in repetitive region (e.g. insertion sequences) were omitted from downstream analyses.

**Whole-genome comparisons and phage analysis.** Whole-genome comparisons were performed using BRIG[56]. Regions of difference were further investigated using the Artemis Comparison Tool (ACT)[57]. A ~25 kb region, missing in all patient 3 strains, was analysed using BLASTn to determine predicted gene content. As the region was predicted to contain phage-like proteins, Ecl1, MS7884 (RBWH patient 1), MS7886 (RBWH patient 2) and MS7890 (RBWH patient 3) draft assemblies were analysed using PHAST[58] to further characterise the phage-like region.

**Pacific Biosciences (PacBio) SMRT sequencing.** A representative *E. hormaechei* isolate from patient 1 (MS7884) was grown on LB agar at 37 °C overnight. Genomic DNA was extracted using UltraClean® Microbial DNA Isolation Kit (MO BIO) as per the manufacturer's instructions. DNA was prepared for sequencing using an 8–12 kb insert library and sequenced on a PacBio RSII sequencer using 1 SMRT cell. MS7884 (RBWH patient 1, isolate 1) was first sequenced on a PacBio RSII sequencer using the P6-C4 sequencing chemistry at the University of Malaya and assembled using the SMRT Analysis suite (version 2.3.0) to give a single chromosome (MS7884B: 4,810,853 bp) and one un-typeable plasmid (pMS7884B: 126,208 bp). However, neither pMS7884B nor the chromosome contained any of the previously identified antibiotic resistance genes, including the carbapenemase *bla*IMP-4 (previously identified using Illumina and PCR). To ensure that the suspected MDR plasmid was retained, we re-extracted genomic DNA from MS7884 grown overnight with 2 µg mL$^{-1}$ meropenem (in 15 mL lysogeny broth, LB). Sequencing was carried out using a different PacBio RSII Sequencer (P6-C4 sequencing chemistry) at the Doherty Institute, University of Melbourne. Assembly of this genome using the SMRT Analysis suite (version 2.3.0) gave an identical chromosome (4,810,853 bp) and a single IncHI2 plasmid (330,060 bp) carrying *bla*IMP-4 (hereafter referred to as pMS7884A). To avoid duplication in the complete genome databases, we have compiled the MS7884A chromosome, pMS7884A plasmid and pMS7884B plasmid as a single-strain MS7884 genome, consistent with our original observation (according to Illumina sequencing) that both plasmids were present in the original patient isolate. Individual PacBio Raw data files are available to enable comparisons of the methylome between the two spontaneously cured MS7884 derivative isolates (Supplementary Tables 6 and 7).

All PacBio SMRT sequences were manually closed using the Artemis Comparison Tool (ACT)[7]. The chromosome and plasmids were polished using both the PacBio reads (with Quiver through the SMRT Analysis suite) and the Illumina reads (using BWA v0.7.5a-r405 with consensus calling through Pilon v1.21) to remove erroneous indels. The final genome was annotated using Prokka (v1.12-beta). The integron regions were manually annotated using a combination of RAC[9], as implemented through MARA (http://app.spokade.com/mara/), and Integrall[10]. Insertion sequences (IS) were manually annotated using ISSaga[11]. Modified bases and associated motifs were detected using the SMRT analysis suite (version 2.3.0) to determine genome-wide methylation.

**Colony polymerase chain reaction (PCR) to identify plasmids in MS7884.** MS7884 was grown on LB agar overnight at 37 °C. Single colonies were resuspended in 50 µL sterile dH$_2$O, boiled for 15 min and centrifuged at $280 \times g$ for 2 min. Then, 5 µL of supernatant containing colony DNA was used in 25 µL OneTaq PCR reactions, as per the manufacturer's recommendations (New England Biolabs Inc.). PCR primers were designed to identify both plasmids (Supplementary Table 8). PCR was run as follows: Annealing 30 s at 55 °C, extension 1 min at 68 °C, denaturing 30 s at 94 °C, 29 cycles.

**Oxford Nanopore MinION sequencing.** Hospital surveillance identified a *bla*IMP-4-positive *E. hormaechei* isolate in October 2017 from a patient in the Haematology ward of RBWH (MS14449). MS14449 was grown overnight on horse blood agar at 37 °C. DNA was extracted using the MoBio UltraClean Microbial DNA isolation kit (as per the manufacturer's instructions). Briefly, 1.5 µg of DNA was prepared using the 1D Genomic DNA by Ligation sequencing kit (SQK-LSK108 version GDE_9002_v108_revT_18Oct2016). The entire library was loaded onto a MIN106 R9.4 flow cell and run for 26 h using MinKNOW version 1.7.14 on a Mac OSX operating system. A subset of 80,000 fast5 files were basecalled using Albacore (v1.1.1). Basecalled reads were filtered for length (minimum 2000 bp) and quality (minimum 10) using Japsa (https://github.com/mdcao/japsa). The remaining 62,093 reads were assembled using Canu (v1.3) at default settings resolving a single chromosome and two plasmids.

**Metagenomic sequencing and analysis of environmental samples.** Swab and water samples from the ICU and Burns Ward were collected in July 2018. DNA was extracted directly from samples using the Qiagen DNeasy Powersoil extraction kit including Biospec Products 0.1 mm diameter glass beads (as per the manufacturer's instructions). Water samples were concentrated using a Whatman Nucleore 0.2 µm polycarbonate 25 mm diameter filter prior to DNA extraction. Sample DNA was sequenced at the Australian Centre for Ecogenomics on an Illumina NextSeq 500. Libraries were prepared according to the manufacturer's protocol using Nextera XT Library Preparation Kit (Illumina #FC-131-1096). The only alteration to the protocol as outlined was the reduction of total reaction volume for processing in 96-well plate format. Library preparation and bead clean-up was run on the Mantis Liquid Handler (Formulatrix) and Epimotion (Eppendorf #5075000301) automated platform. These programs cover "Tagment Genomic DNA" to "Amplify DNA" in the protocol (Mantis-Nextera XT library prep protocol) and "Clean Up Libraries" in the protocol (Epimotion—Library Clean Up protocol). On completion of the library prep protocol, each library was quantified and QC was performed using the Qubit™ dsDNA HS Assay Kit (Invitrogen) and Agilent D5000 HS tapes (#5067- 5592) on the TapeStation 4200 (Agilent # G2991AA) as per the manufacturer's protocol.

Nextera XT libraries were pooled at equimolar amounts of 1 nM per library to create a sequencing pool. The library pool was quantified in triplicates using the Qubit™ dsDNA HS Assay Kit (Invitrogen). Library QC is performed using the Agilent D5000 HS tapes (#5067–5592) on the TapeStation 4200 (Agilent # G2991AA) as per the manufacturer's protocol. The library was prepared for sequencing on the NextSeq500 (Illumina) using NextSeq 500/550 High Output v2 $2 \times 150$ bp paired end chemistry in the Australian Centre for Ecogenomics according to the manufacturer's protocol.

All samples were screened for species using Kraken v1.0[59]. Samples were also screened for resistance genes using srst2 v0.2.0[50] against the ARG-ANNOT database. MinHash sketches of the ST90 *E. hormaechei* chromosome MS7884 (GenBank: CP022532.1) and the associated IncHI2 plasmid pMS7884A (GenBank: CP022533.1) were generated using MASH v1.1.1[45] at default settings. Illumina reads for each sample were screened against our reference sketches using the screen function in MASH. Samples that shared ≥90% of hashes were mapped to the reference sequences. Mapped reads were then parsed using Samtools fastq (v1.9) and de novo assembled using Spades v3.11.1 for MLST analysis using Abricate v0.8 (https://github.com/tseemann/abricate) and nucleotide comparison using FastANI[46], ACT[57] and BRIG[56].

**Reporting summary.** Further information on research design is available in the Nature Research Reporting Summary linked to this article.

## Data availability

Genome data have been deposited under Bioproject PRJNA383436. Illumina sequence read data have been deposited to the sequence read archive (SRA) under the accessions SRR5821451–SRR5821467, SRR8789021 and SRR8789023–SRR8789027 [https://trace.ncbi.nlm.nih.gov/Traces/sra/?study=SRP106560]. Nanopore sequence read data have been deposited to the SRA under the accession SRR8789022, PacBio sequence read data have been deposited to the SRA under the accessions SRR5821468 and SRR5821469. Environmental metagenomic sequence read data have been deposited to the SRA under the accessions SRR8801892–SRR8801897 [https://trace.ncbi.nlm.nih.gov/Traces/study/?acc=SRP106560&o=acc_s%3Aa]. The complete genome of MS7884 has been deposited to Genbank under the accessions CP022532–CP022534 [https://www.ncbi.nlm.nih.gov/assembly/GCA_002237465.1].

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

## Acknowledgements

We thank Krispin Hajkowicz, Trish Hurst and Michelle Doidge (Infectious Diseases Department, RBWH, Brisbane, Australia) for collecting environmental samples used in this study. We also would like to acknowledge Andrew Henderson and Ama Ranasinghe (UQCCR) for their help with MIC testing.

## Author contributions

L.W.R., P.N.A.H., S.A.B. and M.A.S. designed the study. P.N.A.H., D.L.P., E.C., H.E.S., H.B., J.L., A.A. and C.H. carried out clinical aspects of study, collected isolates and undertook clinical microbiology analyses. M.-D.P., J.A.G. and L.W.R. cultured isolates and extracted DNA. L.W.R., N.L.B.Z., M.S.-C. and B.M.F. performed the bioinformatic analysis of initial outbreak isolates. L.W.R. performed the molecular testing of initial outbreak isolates and bioinformatic analysis of all subsequent isolates and metagenomic samples. L.W.R. completed the Nanopore sequencing and analysis. L.W.R. and B.M.F. performed the reference genome assembly and metagenomic analysis. K.-G.C., T.M.C. and W.-F.Y. provided the Pacific Biosciences sequencing data. S.A.B. and P.N.A.H. supervised the study. L.W.R., P.N.A.H. and S.A.B. wrote the original manuscript. All authors assisted in review of the original manuscript.

## Competing interests

The authors declare no competing interests.
