## [Peer Review File · Nature Communications]

Reviewers' comments:

Reviewer #1 (Remarks to the Author):

In this manuscript, Roberts et al. describe a comprehensive genomic analysis that followed an outbreak of carbapenem-resistant *Enterobacter hormaechei* in an ICU/Burn Unit in a tertiary referral hospital in Brisbane, Australia. The investigators use several whole genome sequencing based techniques to characterize the outbreak isolates as well as to place the isolates in the context of local epidemiology. The main findings of this investigation are that the outbreak isolates were clonally related to an isolate collected from the same ward two years prior. Additionally, while carbapenem-resistant Enterobacteriaceae from surrounding hospitals were not clonally related to the outbreak strain, the plasmid associated with the carbapenemase (IMP-4) and the associated plasmid (Inch12) were found in locally circulating strains, which the authors interpreted as suggesting plasmid mediated circulation. Finally, continued surveillance several years after the initial outbreak revealed persistence of the Inch12 plasmid in the hospital, and metagenomic-based environmental surveillance suggested the presence of the outbreak strain in the hospital plumbing. This manuscript describes an interesting genomic-informed outbreak investigation of a concerning multidrug-resistant organism. However, contrary to the author's claims, it is not clear that the study represents a significant advance beyond previous studies applying WGS in similar contexts. The use of WGS for CRE outbreak investigations has been reported extensively (e.g. <https://www.ncbi.nlm.nih.gov/pubmed/22914622> and <https://www.ncbi.nlm.nih.gov/pubmed/27919898>), the use of long-read sequencing to characterize carbapenemase-associated plasmids is a common approach (e.g. <https://www.ncbi.nlm.nih.gov/pubmed/27067320/>, <https://mbio.asm.org/content/9/1/e02011-17>) and evidence of CRE in hospital plumbing has been extensively reported (e.g. <https://www.ncbi.nlm.nih.gov/pubmed/29950189>). Moreover, advances claimed in the first paragraph of the discussion are either unjustified (e.g. sequencing was not actually performed in real-time, sequencing did not actually guide interventions) or have been previously reported (e.g. long-read sequencing to resolve chromosomes/plasmids, nanopore for rapid sequencing and WGS to understand transmission). A potentially novel contribution is the application of metagenomic analyses of the hospital water when traditional culture-based methods failed, but the metagenomic methods are sparsely explained and the support for the resulting conclusions are unconvincing.

Major Comments

1. The metagenomic bioinformatic methods and support for the presence of ST90 in the metagenomic analyses are incompletely described. Moreover, while the authors state that metagenomic reads mapped across the ST90 reference genome and plasmid with high identity, it seems that not even a complete MLST match could be found – so how should these two findings be reconciled? Given the existence of a high quality reference genome, more precise analyses should be able to be performed to quantitatively assess the likelihood that the outbreak strain was present in the metagenome.
2. The clinical case report text in the introduction replicates everything in Figure 1. Either the figure or the text could be minimized. Additionally, text about the clinical cases should be in the Results.
3. In addition to highlighting the importance of CPE, the introduction could be used to present the concept of using multiple WGS-based technologies for clinical/public health outbreak investigations, especially to distinguish clonal vs. mobile genetic element mediated circulation.
4. While an important topic, the paragraph in the discussion about meropenem susceptibility testing discrepancies (page 17 lines 415 to 427) seems sudden and disjointed from the remainder of the manuscript.
5. The discussion alludes to real-time sequencing leading to infection prevention interventions.

This is somewhat unconvincing as first, genomics was not really done in real time and second none of the infection prevention interventions relied solely on genomic information (e.g. environmental surveillance is a logical step when observing a rare resistant organism separated over time in the same unit with no other common exposures).

6. The author's state on line 316 in the results that "Oxford Nanopore MinION sequencing has advanced in recent years to provide long-read sequencing and real-time analysis of bacterial isolates. However, no method has been established for its use in rapidly contextualising new isolates during ongoing outbreaks." However, there is not evidence that the author's present a new method to facilitate real-time genomic surveillance, but rather just perform standard analyses to compare whole genome sequences.

Minor Comments

1. Figure 3 legend references blue and orange but the figure has blue and green

2. In Figure 3, while the text legend states the Ecl1 represents all 2015 RBWH outbreak isolates, it would be helpful to also make this apparent in the visual figure legend (when labeling the pink tip)

Reviewer #2 (Remarks to the Author):

This manuscripts presents the analysis of a CRE outbreak in a hospital using three HTS approaches: WGS with short reads (Illumina) to delimit the similarity of different putative outbreak isolates as belonging to the same *Enterococcus hormaechei* ST90 and with almost identical sequences at the core genome; WGS with long reads (PacBio), to establish plasmid IncHI2 as responsible for the IMP-4 production that was common to all outbreak isolates; and metagenomic sequencing of environmental samples from the plumbing system of the hospital, to identify *E. hormaechei* ST90 and plasmid IncHI2 within the hospital plumbing system.

I have found the manuscript interesting and well written, with most details on the analyses performed described in the Supplementary material, thus making the reading of the main text much easier and more fluid. This represents an excellent example of how to investigate a nosocomial outbreak of MDR bacteria from the identification and delimitation of the outbreak and the affected patients to the tracking of its environmental source in the plumbing system. As far as I know, there is no similar comprehensive application of high throughput sequencing technologies for a nosocomial outbreak of bacteria.

Apart from the originality of their work, the authors provide a detailed account and clear example of the limitations of traditional methods for tracking outbreaks and the advantages provided by using the different techniques of HTS. For instance, the sequencing of the IncHI2 plasmid allowed them to explain the lack of MDR in one patient related to the outbreak but with a deletion encompassing the MDR-determining genes in the plasmid of his infecting strain. Another interesting case is the characterization of the environmental reservoir and the continuous transfer of the IncHI2 plasmid among species, thus explaining its persistence in the hospital system and the occasional appearance of outbreaks.

I think that this paper will have a high impact in the field and, as a consequence, other groups will try to replicate the methods used when facing similar problems. In this context, I would have preferred the authors to use some more updated software to help improve future reproducibility. For instance, the Nasoni program for mapping reads to a reference genome is quite obsolete and the VBC, who produced it, was closed on 2014. A comment on the supplementary material on alternative methods/programs would be advisable.

Overall, I have not found any important details or description of methods missing from the manuscript and the supplementary material and I recommend its acceptance for publication.

Prof. Fernando Gonzalez-Candelas

Reviewer #3 (Remarks to the Author):

This is an excellent example of an outbreak investigation that demonstrates the rational and effective utilization of three different sequencing technologies to investigate an outbreak of carbapenem-resistant *Enterobacter hormaechei*. The paper is well written and the methods used are consistent with routinely used methods for sequencing and bioinformatics analysis. An outbreak of *E. hormaechei* was first noticed in 2015 and whole-genome sequencing was able to reveal carbapenem resistance via an IMP-4 metallo-beta-lactamase encoded on an IncHI2 plasmid. Subsequent surveillance suggested the persistence of this strain and metagenomic sequencing of environmental samples was able to detect the presence of the same organism and plasmid in the hospital plumbing system.

At first read, one might think this was yet another paper showing off the unwarranted use of next-generation sequencing technologies for simple outbreaks that could have been resolved by fingerprinting methods. However, the authors should be commended for choosing the most appropriate technology at every stage of the investigation.

Major Comments:

The discrepancy between the E-test and Vitek-2 results for meropenem are concerning. This is not the experience for many other clinical labs. Was there a problem with the E-test strips? Did the controls for the E-test turn out correct? Was there a "phantom zone" and possible misinterpretation of the MIC? Could they send the isolates to a reference laboratory to repeat the E-test? If the E-test is truly undercalling the meropenem result on certain organisms or for certain carbapenemases, then it needs further investigation and should be reported to alert other clinical labs.

Contamination is a serious issue in metagenomic sequencing. From the manuscript, it seems that the WGS was performed in 2015 and the metagenomic sequencing was performed in 2018. Some labs are starting to dedicate one sequencer for metagenomic samples only and a second sequencer only for bacterial whole-genome sequencing. As only a small proportion of matching reads were detected by metagenomic sequencing, can the authors provide re-assurance that these are not contaminants? i.e., were there any *Enterobacter hormaechei* samples sequenced around the same time as the metagenomic samples?

Was the *Enterobacter* recovered from the ETT samples from patients 1 and 2 considered to be causing VAP or a colonizer? Is there any evidence to support their conclusion - such as Gram stain or clinical symptoms?

Minor Comments:

1. Gram stain is capitalized, but gram-negative should not be
2. intensive care unit, burns unit, hematology ward do not need to be capitalized unless it is part of a formally named unit
3. line 96 - Patient 1 and Patient 2 - not consistently capitalized compared to rest of text
4. line 202 - manufacturer's
5. line 259 - refer "to" supplementary appendix
6. line 334 - "real-time" PCR is not capitalized
7. line 335 - "similarity" should be replaced by "identity"
8. line 417 - "Vitek2" missing space or dash
9. Table 1 - the "1" asterisk (under Vitek2 and in the footnote) may not be required
10. Combine Figures 1 and 2. The value of including the antibiotic treatments is not clear in the context of this paper. If those were removed, the two figures could easily be combined.

Response to reviewers NCOMM-19-12498:

Please find responses to reviewer comments below (in blue text). Yellow highlight is used to denote major changes to manuscript in this response letter and in main document. Minor changes and corrections relative to our original submission are shown in 'track changes' the body of the main document and supplementary appendix.

Reviewer #1 (Remarks to the Author):

In this manuscript, Roberts et al. describe a comprehensive genomic analysis that followed an outbreak of carbapenem-resistant *Enterobacter hormaechei* in an ICU/Burn Unit in a tertiary referral hospital in Brisbane, Australia. The investigators use several whole genome sequencing based techniques to characterize the outbreak isolates as well as to place the isolates in the context of local epidemiology. The main findings of this investigation are that the outbreak isolates were clonally related to an isolate collected from the same ward two years prior. Additionally, while carbapenem-resistant Enterobacteriaceae from surrounding hospitals were not clonally related to the outbreak strain, the plasmid associated with the carbapenemase (IMP-4) and the associated plasmid (IncH12) were found in locally circulating strains, which the authors interpreted as suggesting plasmid mediated circulation. Finally, continued surveillance several years after the initial outbreak revealed persistence of the IncH12 plasmid in the hospital, and metagenomic-based environmental surveillance suggested the presence of the outbreak strain in the hospital plumbing. This manuscript describes an interesting genomic-informed outbreak investigation of a concerning multidrug-resistant organism.

However, contrary to the author's claims, it is not clear that the study represents a significant advance beyond previous studies applying WGS in similar contexts. The use of WGS for CRE outbreak investigations has been reported on extensively (e.g. <https://www.ncbi.nlm.nih.gov/pubmed/22914622> and <https://www.ncbi.nlm.nih.gov/pubmed/27919898>), the use of long-read sequencing to characterize carbapenemase-associated plasmids is a common approach (e.g. <https://www.ncbi.nlm.nih.gov/pubmed/27067320/>, <https://mbio.asm.org/content/9/1/e02011-17>) and evidence of CRE in hospital plumbing has been extensively reported (e.g. <https://www.ncbi.nlm.nih.gov/pubmed/29950189>). Moreover, advances claimed in the first paragraph of the discussion are either unjustified (e.g. sequencing was not actually performed in real-time, sequencing did not actually guide interventions) or have been previously reported (e.g. long-read sequencing to resolve chromosomes/plasmids, nanopore for rapid sequencing and WGS to understand transmission). A potentially novel contribution is the application of metagenomic analyses of the hospital water when traditional culture-based methods failed, but the metagenomic methods are sparsely explained and the support for the resulting conclusions are unconvincing.

We thank the reviewer for their constructive criticism that we have addressed in depth below.

Major Comments

1. The metagenomic bioinformatic methods and support for the presence of ST90 in the metagenomic analyses are incompletely described. Moreover, while the authors state that metagenomic reads mapped across the ST90 reference genome and plasmid with high identity, it seems that not even a complete MLST match could be found – so how should these two findings be reconciled? Given the existence of a high quality reference genome, more precise analyses should be able to be performed to quantitatively assess the likelihood that the outbreak strain was present in the metagenome.

We accept that the metagenomic analysis could be more clearly presented. Supplementary Dataset 2 now contains both the proportion of reads mapping from each original sample, and the average nucleotide identity (ANI) of our metagenome-assembled genomes (MAGs) to our reference sequences. “High confidence matches” had >90% ANI across more than 95% of the reference when comparing the MAGs to our reference sequences. “Low confidence matches” were between 80-90% ANI across 55-80% of the reference sequence. We have changed the main text to better reflect these more robust methods that were used to support the identification of the outbreak clone within these samples (see below).

We also thank the reviewer for highlighting the apparent incongruity of having a high quality match to the metagenomic assemblies but inconclusive *in silico* MLST results. In fact, because a single SNP can result in an incorrect MLST allele and the ANI was 93-97% (corresponding to 3-7 nucleotide differences in every 100 bp window) we were a little surprised how many of the seven individual house-keeping genes were 100% conserved. In the original submission we noted that some MLST alleles were incomplete, which may have led to the confusion. In fact, most alleles were present and matched the outbreak strain in samples R5514 and R5537 (4 and 5 of 7 MLST alleles were matched with our reference *E. hormaechei* ST90, respectively). As expected, other alleles had some nucleotide ambiguities due to the inherent variability of read coverage within a mixed environmental sample. In the revised manuscript we now clarified the level of identity between our MAGs and the complete reference genome (with ANI), and state that almost identical ST90 alleles were identified from the MAGs (see below). Note that all MLST allele data for the metagenome samples are presented in Table S8:

Page 11-12, Line 309-311:

“Nucleotide comparison of the metagenomic assembled genomes (MAGs) for these samples (R5514 and R5537, both taken from floor drains) to our *E. hormachei* reference genome revealed a high level of nucleotide identity across the entirety of the chromosome and plasmid, equivalent to 93-97% average nucleotide identity (ANI) across >95% of the reference sequences (Supplementary Dataset 2 and Supplementary Figures 12 and 13). MLST analysis of both MAGs was also able to detect an almost identical set of alleles for ST90 (Table S8).”

2. The clinical case report text in the introduction replicates everything in Figure 1. Either the figure or the text could be minimized. Additionally, text about the clinical cases should be in the Results.

We thank reviewer 1 for the comment, however we believe the clinical case report is an important feature of the introduction. The results of the manuscript pertain to the isolates themselves and the wider implication of endemic *blaIMP-4*. The clinical case report describes only the patients and provides context to the isolates, and as such has been included as introductory material.

Both the clinical case report and Figure 1 are complementary, as the report provides a detailed description of the clinical outcomes, while Figure 1 enables the reader to immediately assess the overall timeline of the three patients (and the outbreak investigation). The clinical co-authors on this manuscript believe this technology and the described report have major clinical ramifications and hence we respectfully have not minimized the text.

3. In addition to highlighting the importance of CPE, the introduction could be used to present the concept of using multiple WGS-based technologies for clinical/public health outbreak investigations, especially to distinguish clonal vs. mobile genetic element mediated circulation.

We agree and have added a short paragraph to the introduction section outlining the use of WGS in clinical/public health outbreaks:

Page 4-5, Line 118-125

“Within the last decade, whole genome sequencing (WGS) has become more accessible and affordable, resulting in its increased use in many fields, including clinical microbiology. One of the main features of using WGS to characterise clinically-relevant bacteria comes from its ability to provide strain relatedness at the resolution of a single nucleotide. To date, many researchers globally have utilised this technique to understand transmission within hospital settings beyond what can be determined using traditional culture-based diagnostics alone. Examples include the expansion of a single clone¹⁷⁻¹⁹ and less commonly the expansion of a mobile genetic element (MGE)^{20,21}. However, many of these studies were conducted retrospectively and focused primarily on a single sequencing technology.”

4. While an important topic, the paragraph in the discussion about meropenem susceptibility testing discrepancies (page 17 lines 415 to 427) seems sudden and disjointed from the remainder of the manuscript.

We acknowledge that this section may seem unexpected as it is separate to the primary outcomes of this manuscript. However, we anticipate that this paper will interest a broad audience, including clinicians, who will likely be very interested in this discrepancy with meropenem sensitivity testing. This has already been demonstrated by Reviewer 3's feedback. We certainly considered moving this section elsewhere in the discussion, however, on reflection we believe that its current position is the most appropriate place.

5. The discussion alludes to real-time sequencing leading to infection prevention interventions. This is somewhat unconvincing as first, genomics was not really done in real time and second none of the infection prevention interventions relied solely on genomic information (e.g. environmental surveillance is a logical step when observing a rare resistant organism separated over time in the same unit with no other common exposures).

We agree with reviewer 1 that we have not made it clear how our analyses constitute “real-time” WGS based on the initial outbreak response. We note that WGS-led outbreak investigations had not been implemented in this hospital (or any other in Queensland) prior to our study, so there was no existing infrastructure to support increased turn-around-time needed for true “real-time” WGS as we know it today. Never-the-less, as outlined in Figure 1, our initial WGS in 2015 was performed during the outbreak, days after the phenotypic identification of suspected outbreak isolates from patient 3. At that time it was unknown whether there would be more cases in additional patients (fortunately there were not). Critically it was WGS that unequivocally linked the 2015 outbreak to a hospital source (through comparisons with public genome data).

In the following year we implemented real-time WGS for an unrelated outbreak at the same ward that identified the persistence of the *E. hormaechei* outbreak strain. In 2017 we undertook real-time sequencing using the Oxford Nanopore MinION instrument to rule out a suspected spread of the outbreak to another ward. In all cases we believe our response was as close to “real-time” as possible, and certainly can be considered “rapid”. However, we acknowledge that the use of the phrase “real-time” could be misleading to some readers. Accordingly we have only retained “real-time” for the Nanopore sequencing (Line 279) and have modified the introduction and conclusion to instead highlight the integration of multiple technologies, which is a key novelty in our study (as pointed out by Reviewer 2):

Page 5, Line 127:

“...we describe the integration ~~real-time use~~ of several WGS technologies...”

Page 18, Line 414:

“Real-time Rapid application of this technology revealed an unexpected clonal relationship...”

Reviewer 1 rightly queries the extent to which infection prevention interventions were driven mainly by WGS results. Firstly, the main contribution of WGS to the infection control response was the confirmation provided by unambiguously linking the isolates from the three patients and an isolate from 2013. This crucial link to a 2013 isolate, which would not have been recognised without WGS, ultimately identified a probable environmental source that was persistent within the hospital. This led to extensive environmental sampling and continued surveillance that was beyond the standard infection control response. Secondly, the SNV profiles from the WGS data enabled us to rule out transmission between patient 2 and patient 3 (Page 8, Line 224-226). Finally, the ability of WGS to rule out a subsequent *E. hormaechei* isolated from a Hematology ward from being related to the outbreak enabled infection control to limit their response, ultimately reducing costs relating to additional screening and cleaning. The role of WGS in the infection control response is outlined in the results section entitled “Integration of WGS with infection control response” (Pages 8-9, Line 228-238).

6. The author’s state on line 316 in the results that “Oxford Nanopore MinION sequencing has advanced in recent years to provide long-read sequencing and real-time analysis of bacterial isolates. However, no method has been established for its use in rapidly contextualising new isolates during ongoing outbreaks.” However, there is not evidence that the author’s present a new method to facilitate real-time genomic surveillance, but rather just perform standard analyses to compare whole genome sequences.

We thank Reviewer 1 for their comment, and agree that our approach does not strictly constitute a new method *per se*. Our approach adapts existing methods to demonstrate the capability of Nanopore MinION data for outbreak investigation which is still very much an emerging field. It is well established that long-read sequencing technologies, such as Nanopore, suffer from high error rates in single reads. This high error rate has led a perception that Nanopore MinION sequencing may be unsuitable for bacterial outbreak analysis, as distinguishing closely related strains requires highly accurate sequencing data. However, determining isolate relationships at the strain level using only a Nanopore instrument would drastically reduce the turn-around time for reporting a new case as either related or unrelated to the ongoing outbreak (this could potentially reduce the turn-around time from roughly 1 week for Illumina, to less than 2 days for Nanopore). In fact, an early demonstration of Nanopore MinION real-time capabilities (Reference 19: Quick et al., Genome Biology (2015)) took a similar approach to the problem using reference-guided assembly of data streamed at 10 minute intervals. However, such approaches are still yet to be widely established and we believe that it is important to demonstrate to the community that using Nanopore data alone is a feasible approach in an outbreak context. We have now modified the text to clarify that our approach is simply demonstration of using Nanopore data alone and have removed the claim that this is a novel

method. We have also provided further details on our approach and results in the Supplementary appendix to improve reproducibility. Furthermore, we have included a citation to Quick et al (Ref 19) to ensure that this earlier work is recognised:

Page 10-11, Line 278-288:

“In 2017, another *bla*_{IMP-4} positive *E. hormaechei* (MS14449) was isolated from the hematology ward. Oxford Nanopore MinION sequencing has advanced in recent years to provide long-read sequencing and real-time analysis of bacterial isolates¹⁹. However, it remains a challenge to rapidly contextualise new isolates during an outbreak, in part due to the relatively high single-read error rate of long-read data. These errors, which are not easily differentiated from true differences, make it difficult to determine the precise genetic relationship between new isolates and the outbreak strain. However, we found that Nanopore data alone was sufficient to rule out MS14449 from the ongoing outbreak within two days of receiving the isolate (see supplementary appendix). By contextualising the *de novo* Nanopore assembly with genomes of publicly available *E. cloacae* complex strains and existing outbreak isolates, we determined that the *E. hormaechei* isolated was not related to the 2015 outbreak isolates, but did carry a near identical IncHI2 plasmid (Figure S7-S9).”

Reference 19: Quick, J. *et al.* Rapid draft sequencing and real-time nanopore sequencing in a hospital outbreak of Salmonella. *Genome Biol* **16**, 114, doi:10.1186/s13059-015-0677-2 (2015).

Minor Comments

1. Figure 3 legend references blue and orange but the figure has blue and green

We thank Reviewer 1 for noticing this error and have edited the legend accordingly.

2. In Figure 3, while the text legend states the Ec11 represents all 2015 RBWH outbreak isolates, it would be helpful to also make this apparent in the visual figure legend (when labeling the pink tip)

We accept Reviewer 1’s suggestion and have edited the figure accordingly.

Reviewer #2 (Remarks to the Author):

This manuscript presents the analysis of a CRE outbreak in a hospital using three HTS approaches: WGS with short reads (Illumina) to delimit the similarity of different putative outbreak isolates as belonging to the same *Enterococcus hormaechei* ST90 and with almost identical sequences at the core genome; WGS with long reads (PacBio), to establish plasmid

IncHI2 as responsible for the IMP-4 production that was common to all outbreak isolates; and metagenomic sequencing of environmental samples from the plumbing system of the hospital, to identify *E. hormaechei* ST90 and plasmid IncHI2 within the hospital plumbing system.

I have found the manuscript interesting and well written, with most details on the analyses performed described in the Supplementary material, thus making the reading of the main text much easier and more fluid. This represents an excellent example of how to investigate a nosocomial outbreak of MDR bacteria from the identification and delimitation of the outbreak and the affected patients to the tracking of its environmental source in the plumbing system. As far as I know, there is no similar comprehensive application of high throughput sequencing technologies for a nosocomial outbreak of bacteria.

Apart from the originality of their work, the authors provide a detailed account and clear example of the limitations of traditional methods for tracking outbreaks and the advantages provided by using the different techniques of HTS. For instance, the sequencing of the IncHI2 plasmid allowed them to explain the lack of MDR in one patient related to the outbreak but with a deletion encompassing the MDR-determining genes in the plasmid of his infecting strain. Another interesting case is the characterization of the environmental reservoir and the continuous transfer of the IncHI2 plasmid among species, thus explaining its persistence in the hospital system and the occasional appearance of outbreaks.

I think that this paper will have a high impact in the field and, as a consequence, other groups will try to replicate the methods used when facing similar problems. In this context, I would have preferred the authors to use some more updated software to help improve future reproducibility. For instance, the Nsoni program for mapping reads to a reference genome is quite obsolete and the VBC, who produced it, was closed on 2014. A comment on the supplementary material on alternative methods/programs would be advisable.

Overall, I have not found any important details or description of methods missing from the manuscript and the supplementary material and I recommend its acceptance for publication.

Prof. Fernando Gonzalez-Candelas

We thank reviewer 2 for their comments and are very grateful for their positive assessment of our research.

We acknowledge that the use of Nsoni is uncommon now but it is also important to note that we carried out the initial sequencing and analysis in 2015 at which time Nsoni was more popular. Nsoni itself is a software toolkit (written in Perl) that aligns reads to a reference using a read-mapper (SHRiMP or Bowtie) and then calls variants in the read-pileup using other well-known software (Freebayes and Picard) and custom filtering. We acknowledge that the SHRiMP read-mapper itself is no longer maintained and an alternative base-calling method would be recommended if we began our study today. However, at the time, all SNVs were manually inspected at the read level before SNVs were reported.

To address any concerns over reproducibility we have included an additional section in the Supplementary appendix that outlines the consistency of SNV calls between Neson1 (implementing either SHRiMP or Bowtie), and Snippy, which is a popular modern alternative to Neson1 from the same author (<https://github.com/tseemann/snippy>). We also note that no additional SNVs were identified by alternative software.

In our revised supplementary appendix we emphasize that the phenotypic differences observed during laboratory growth of some isolates appear to be consistent with the expected consequences of missense or nonsense mutations. Although it is beyond the scope of the present study, we wanted to highlight that within-host SNVs can potentially inform our understanding of pathogen evolution.

Page 18, Line 475:

SHRiMP v2.2.3²⁷ as implemented in Neson1 v0.130²⁸ under default settings was used to determine core single nucleotide polymorphisms (SNPs) between the ten 2015 RBWH *E. hormaechei* genomes to the reference Ecl1 and create a minimum spanning tree (see supplementary appendix for details on use of alternative software). Further details of the Ecl1 assembly and SNP-calling process are provided in the supplementary appendix.

Page 11, Supplementary Appendix, new results section:

“Alternative software for single nucleotide variation (SNV) analysis

Neson1 is a genomic toolkit developed by the Victorian Bioinformatics Consortium (<https://github.com/Victorian-Bioinformatics-Consortium/neson1>) that implements read mappers and variant callers to determine SNVs (and their consequences) given sequence read datasets and a reference genome. Although in common use at the time of the initial outbreak investigation Neson1 and the SHRiMP read-mapper are no longer maintained. The original SNV calls were manually checked at the read level and were supported by observed phenotypic differences between isolates MS7889, MS7892 and MS7893 and the other *E. hormaechei* isolates, which appeared to be consistent with their respective SNV profiles (Supplementary dataset S1, table “snp_consequence” and next section of supplementary appendix).

To ensure that others can reproduce our variant calls we carried out additional read-mapping analysis with alternative software and the complete genome of *E. hormaechei* MS7884A. We found that Neson1 identified the same 4 SNVs regardless of whether SHRiMP or a recent version of Bowtie (v2.3.4.2) was implemented. We also found the same variants using the popular Snippy tool (version 4.4.0), which implements the Burrows-Wheeler Aligner (BWA) and is developed by the original author of Neson1 (<https://github.com/tseemann/snippy>). Finally, we note that the discrepancies between SNV calls in the MS7884A complete genome (Supplementary dataset S1) and the original SNVs presented in Figure 2 are due to reverse complementation of some sequence in the original draft reference genome of strain Ecl1 relative to the final assembly of the MS7884A chromosome.”

Reviewer #3 (Remarks to the Author):

This is an excellent example of an outbreak investigation that demonstrates the rational and effective utilization of three different sequencing technologies to investigate an outbreak of carbapenem-resistant *Enterobacter hormaechei*. The paper is well written and the methods used are consistent with routinely used methods for sequencing and bioinformatics analysis. An outbreak of *E. hormaechei* was first noticed in 2015 and whole-genome sequencing was able to reveal carbapenem resistance via an IMP-4 metallo-beta-lactamase encoded on an IncHI2 plasmid. Subsequent surveillance suggested the persistence of this strain and metagenomic sequencing of environmental samples was able to detect the presence of the same organism and plasmid in the hospital plumbing system.

At first read, one might think this was yet another paper showing off the unwarranted use of next-generation sequencing technologies for simple outbreaks that could have been resolved by fingerprinting methods. However, the authors should be commended for choosing the most appropriate technology at every stage of the investigation.

We thank the reviewer for their positive comments.

Major Comments:

The discrepancy between the E-test and Vitek-2 results for meropenem are concerning. This is not the experience for many other clinical labs.

We agree that this result should be clarified and have addressed specific questions below. In response to the reviewer's comments we carried out additional susceptibility tests with broth microdilution (BMD) and found that the meropenem MICs fell below the clinical breakpoints, thus suggesting that the Vitek 2 is "over-calling" resistance. Overall, the BMD results were closer to the results generated with the E-test.

Was there a problem with the E-test strips?

No. The E-tests were measured by experienced technicians in a National Association of Testing Authorities, Australia (<https://www.nata.com.au>, NATA) accredited laboratory (Accreditation # 2639, Site # 2632), according to the manufacturer's instructions.

Did the controls for the E-test turn out correct?

Yes. All quality control organisms were within range and all E-tests were within date. The E-test freezer is continuously monitored for temperature fluctuations and no temperature anomalies were noted.

Was there a "phantom zone" and possible misinterpretation of the MIC?

No phantom zone was observed. E-tests were read by multiple trained and competent technical staff and confirmed with a supervisor.

Could they send the isolates to a reference laboratory to repeat the E-test?

The E-tests were carried out at a reference laboratory. All testing occurred at the Pathology Queensland central laboratory which is the state reference laboratory for Queensland. Tests were overseen by senior Pathology Queensland staff including co-author Dr Haakon Bergh, who maintains a state-wide 'Microbiology Reference Culture Collection' of over 400 bacteria and fungi for 34 laboratories in accordance with NATA and the Australian Government Therapeutic Goods Administration (<https://www.tga.gov.au>, TGA).

If the E-test is truly undercalling the meropenem result on certain organisms or for certain carbapenemases, then it needs further investigation and should be reported to alert other clinical labs.

In our opinion it is more likely that the Vitek 2 instrument is overcalling the MICs for IMP-4 producers: the E-test and BMD results were consistent. This is a recognised phenomenon with our locally dominant CPE (specifically, *Enterobacter* carrying *bla*_{IMP4}) and the workflow

established across the statewide laboratory network only reports the meropenem E-test MICs for guiding clinical therapy. Discrepancies between meropenem MIC values determined by different methods have also been well recognised by others for CPE, e.g. <https://doi.org/10.1128/JCM.00267-10> (for *Klebsiella pneumoniae*) and <https://doi.org/10.1093/jac/dky276> (for a range of *Enterobacteriaceae*). In the latter article, Haldorsen et al. wrote that:

“Using BMD as the reference method, we noted an overall higher EA and CA with gradient tests compared with semi-automated AST. Furthermore, we observed an overestimation of meropenem MIC for CPE when using semi-automated systems compared with BMD, resulting in high ME rates for class A and class B carbapenemase-producing strains. The tendency of semi-automated systems to overcall meropenem resistance in CPE might have the undesirable effect of deterring appropriate use of carbapenems in the treatment of infections caused by MDR *Enterobacteriaceae* with limited therapeutic options.”

Contamination is a serious issue in metagenomic sequencing. From the manuscript, it seems that the WGS was performed in 2015 and the metagenomic sequencing was performed in 2018. Some labs are starting to dedicate one sequencer for metagenomic samples only and a second sequencer only for bacterial whole-genome sequencing. As only a small proportion of matching reads were detected by metagenomic sequencing, can the authors provide re-assurance that these are not contaminants? i.e., were there any *Enterobacter hormaechei* samples sequenced around the same time as the metagenomic samples?

We thank Reviewer 3 for this insight, and agree that contamination can pose a problem in these circumstances, however, we do not believe this is the case here. We present the following as reassurance that these results reflect true presence and not sequencing contamination:

- (i) All metagenomic sequencing runs included two negative controls, which also did not contain sequence reads matching *E. hormaechei* or the IncHI2 plasmid.
- (ii) All metagenomic sequencing was carried out by the Australian Centre for Ecogenomics sequencing service, which specialises in high-throughput Illumina sequencing of metagenomic samples and serves the wider community beyond our institution.
- (iii) The extraction of DNA from *E. hormaechei* isolates and the hospital environmental samples were carried out in two different laboratories (years apart, as noted by the reviewer). If there was contamination, we would expect to see it more broadly across all samples.
- (iv) From a total of 50 swab and water samples, we only detected the presence of *E. hormaechei* and/or the IncHI2 plasmid in 5 samples. At least two of these samples

had high proportions of sequence reads matching our reference, not simply trace amounts.

Based on this evidence, we are confident that the detection of our strain/plasmid of interest is not contamination but true presence in the metagenomic sample.

We have updated our manuscript to address the reviewers comments and provide more information about the metagenome datasets. We also note that in some samples there were sufficient reads to assemble near-complete genomes (metagenome assembled genomes or MAGs), and have provided the actual numbers of reads that map to our reference sequences in supplementary material.

Was the Enterobacter recovered from the ETT samples from patients 1 and 2 considered to be causing VAP or a colonizer? Is there any evidence to support their conclusion - such as Gram stain or clinical symptoms?

The diagnosis of ventilator associated pneumoniae (VAP) is contentious (Stevens JP et al Crit Care Med. 2014 Mar;42(3):497-503. doi: 10.1097/CCM.0b013e3182a66903.) At the time of isolation, our clinical co-authors thought both cases reflected colonisation, not infection as other systemic signals of infection were not present. Hence in both cases antibiotics were not initially prescribed (see below for relevant patient history).

Patient 1: The *E. cloacae* (now known to be *E. hormachei*) was present only as scant growth mixed with normal respiratory flora, with occasional white cells and epithelial cells on direct Gram staining, with no gram negative bacilli. This was thought to represent colonisation, there was no significant change in the of the patient at the time, and the patient was not treated with antibiotics.

Patient 2: Similarly, the *E. cloacae* (now known to be *E. hormachei*) was initially present only as scant growth in the ET aspirate, with no organisms or while cells seen on gram stain, with scant epithelial cells. The patient had received a single dose of piperacillin-tazobactam 2 days prior to the sample being collected but the culture was considered to represent colonisation. However, Patient 2 subsequently developed signs of sepsis over subsequent days, and the same organism was grown from blood and urine cultures (see Figure 1), which prompted antibiotic treatments.

1. Gram stain is capitalized, but gram-negative should not be - corrected throughout
2. intensive care unit, burns unit, hematology ward do not need to be capitalized unless it is part of a formally named unit - changed throughout
3. line 96 - Patient 1 and Patient 2 - not consistently capitalized compared to rest of text - corrected throughout

4. line 202 - manufacturer's - corrected
5. line 259 - refer "to" supplementary appendix - corrected
6. line 334 - "real-time" PCR is not capitalized - corrected
7. line 335 - "similarity" should be replaced by "identity" - corrected
8. line 417 - "Vitek2" missing space or dash - corrected
9. Table 1 - the "1" asterisk (under Vitek2 and in the footnote) may not be required - changed as per suggestion
10. Combine Figures 1 and 2. The value of including the antibiotic treatments is not clear in the context of this paper. If those were removed, the two figures could easily be combined.

We respectfully disagree. As demonstrated in our response to the ETT question above, we believe that it is of benefit to fully describe the antibiotic treatment regime, especially given the expected interest from the clinical scientific community.